# How Many Students and Items Are Optimal for Teaching Level Evaluation of College Teachers? Evidence from Generalizability Theory and Lagrange Multiplier

Guangming Li [1,2]

[1] Key Laboratory of Brain, Cognition and Education Sciences, Ministry of Education, South China Normal University, Guangzhou 510631, China; lgm2004100@m.scnu.edu.cn

[2] School of Psychology, Center for Studies of Psychological Application and Guangdong Key Laboratory of Mental Health and Cognitive Science, South China Normal University, Guangzhou 510631, China

**Abstract:** Budget and cost are two of the problems that cannot be ignored when conducting a measure study. Based on the application of generalizability theory, combined with Lagrange multiplier, this paper explores how many students and items are optimal for teaching level evaluation of college teachers under budget constraints to maintain the sustainable development of higher education. A total of 397 students are required to evaluate 10 teachers' teaching level using the Teaching Level Evaluation Questionnaire for College Teachers, and we make different generalizability designs (i.e., $(s:t) \times i$, $(s:t) \times (i:v)$ and $(s:t) \times (i:v) \times o$) for the collected data. The study unifies the Lagrange multiplier formula, derives the optimal sample size formula of different designs under budget constraints in generalizability theory, and calculates the optimal sample size for teaching level evaluation of college teachers in different designs with the estimated variance components. Results indicate that: (1) the unified formula of Lagrange multiplier has a stronger robustness and can be applied to different study designs under budget constraints in generalizability theory; (2) the occasion has a great effect on teaching level evaluation for college teachers; (3) the $(s:t) \times (i:v) \times o$ design has a high efficiency in estimating the optimal sample size of teaching level evaluation for college teachers; (4) the design of $(s:t) \times (i:v) \times o$ is the optimal generalizability design of teaching level evaluation for college teachers under budget constraints in generalizability theory; and (5) under budget constraints of teaching level evaluation for college teachers in generalizability theory, the optimal sample size of students is 31 for each teacher and the optimal sample size of items is 7 for each dimension.

**Keywords:** generalizability theory; Lagrange multiplier; budget constraints; estimating the optimal sample size; teaching level evaluation

## 1. Introduction

The teaching level evaluation for college teachers is an important basis for assessments and promotions of relevant teachers by colleges (Spooren et al., 2014) [1]. At present, the focus of higher education assessment turns into the assessment of students' learning outcomes (Le and Xin, 2015) [2]. In the evaluation of teaching level of college teachers, the "students' evaluation of teaching" has become an important part of the evaluation of teaching qualities in most colleges (Le and Xin, 2015; Meyer et al., 2014; Wolbing and Riordan, 2016) [2–4]. Usually, students evaluate the teacher on several indicator dimensions using the teaching level evaluation questionnaires assigned by colleges, and the average score of the evaluation is used to express the teaching level of college teachers. However, the current practice still has some problems.

It is difficult to clarify the complicated relationship among existing influencing factors in the evaluation of the teaching level of college teachers. Although the students are usually seen as the subject of evaluation, it is not appropriate for most colleges only to take the students as the main influencing factor. There are some reasons, as follows: on the one hand,

there are several other factors (such as evaluation items, evaluation occasions, evaluation courses, and evaluation majors, etc.) influencing the evaluation in addition to the main factor students (Chang and Hocevar, 2000; Yang and Chang, 2003) [5,6]; on the other hand, the relationship among them will be complicated if all factors are taken into account in the evaluation, and there are also various combinations of corresponding factors.

Less consideration of budget constraints in the evaluation process results in less cost consideration. In general, the reliability of the evaluation will also increase or decrease as the number of the students who assess their teachers increases or decreases (Lakes, 2013) [7]. However, it is necessary to consider the budget and cost in the process of evaluating the teaching level of college teachers when conducting survey research (Hill et al., 2012; Meyer et al., 2014) [3,8]. In fact, the researchers should consider how to design a measurement program with relatively high feasibility and reliability under budget constraints in designing research procedures (Goldstein and Marcoulides, 1991, 1991) [9,10], because the cost will also be higher when the number of students evaluating teachers is larger. Generally speaking, increasing the number of observations on facets of measurement can reduce the measurement error; thereby improving the reliability of the evaluation when evaluating the teaching level of college teachers. However, when the number of observations increases, the cost for evaluation will undoubtedly increase. Therefore, this is a dilemma for the evaluation. In fact, it is common for all students to participate in the teacher's evaluation in existing colleges with little cost consideration, leading to a serious waste of costs.

It is difficult to estimate the optimal sample size because there is no discussion about the optimal design of the evaluation. Specifically, there are no conclusions on the optimal design plan or the factors that need to be considered to explore the estimation of the optimal sample size at present. Most colleges usually take two methods to restrict: first, for the evaluation of the design program, they often distribute the questionnaire to students online or on-site to fill at the end of the semester, and then collect the data back to evaluate the teacher. Generally, they carry out the optimal program design according to usual practice without additional consideration. Second, for the evaluation of the number of students, most of them make the estimation through artificial regulations. For example, it is often assumed that the optimal number of students in a student evaluation is a sample of a natural class, typically around 30 (Gitomer et al., 2014; Casabianca et al., 2015) [11,12], or, roughly, if the number of students in a course is less than 10, the course will not be included in the evaluation score. However, there is no scientific reason for these practices. In general, the optimal number of students estimated based on people's general qualitative or empirical knowledge is difficult to match with the actual situation and is meaningless, because there are many factors influencing the evaluation of the teaching level of college teachers (Bergsman et al., 2015) [13].

The generalizability theory is one of the modern psychological and educational measurement theories (Qi et al., 2002) [14], which can solve the above problems in the evaluation. The generalizability theory can examine multiple evaluation factors together and analyze the relationship and importance among them. It is a kind of decision theory for the generalizability theory to explore the sample size of different designs under budget constraints. The generalizability theory can explore the optimal number of students, items, and other issues of teaching level of college teachers under budget constraints based on a variety of influencing factors.

The generalizability theory has some advantages on analysis of the teaching level evaluation of college teachers compared with the classical test theory (CTT). The generalizability theory can construct different generalizability designs to carry out the reliability analysis of various factors according to different situations. The generalizability theory can examine the optimal sample size of different generalizability designs under budget constraints based on the relative decision making of various results. It is possible for the generalizability theory to find out which generalizability design scheme is optimal by comparison under certain budgetary constraints.

Corresponding to the above problems, in order to solve the problem of estimating the optimal sample size of the teaching level of college teachers under budget constraints, the generalizability theory usually takes some approaches, as follows.

First of all, different generalizability designs are built. The generalizability theory can take into account these influencing factors together (e.g., students, classes, courses, and majors, etc.) in the evaluation of the teaching level of college teachers, and the object of measurement is the actual teaching level of college teachers. Examining the influence of these factors on the teaching level of college teachers can help to accurately evaluate whether or not a teacher's teaching effect is good, and the level is high. It also can help to better understand the overall teaching level of a teacher from multiple aspects and compare the differences among the teaching levels of different teachers fully. There are also other factors affecting the evaluation of college teachers' teaching level (Chen et al., 2015; Maulana et al., 2015; Oghazi, 2015; Pleschová and Mcalpine, 2016) [15–18]. It is necessary to pay attention to these influencing factors on the objects of measurement and constructing different generalizability designs to reflect the relationship among them in the analysis process using the generalizability theory. Otherwise, these influencing factors may become hidden facets; thereby exaggerating the generalizability coefficient of the evaluation (Brennan, 2001) [19]. For example, a generalizability design of ($s{:}t$) × $i$, ($s{:}t$) × ($i{:}v$) and ($s{:}t$) × ($i{:}v$) × $o$) can present the relationship among these influencing factors. The generalizability theory can construct and analyze different generalizability designs in favor of situational relationships, which show strong advantages. It can also examine the effect of factors in different designs. If the effects are small, they will be ignored. If the effects are large, they are the main factors.

Secondly, the budget and cost of the evaluation are considered. The generalizability coefficient will generally increase until reaching the intended value with one side level increases in the generalizability theory. However, the researchers should consider whether or not to change the research design once there are constraints (e.g., budget constraints from human, material, and financial resource, etc.). In some cases, to increase the generalizability coefficient by 0.01, a larger number of observations of a certain side are required, which may require more money than budgeted. When this happens, the researchers should consider whether or not to increase the number of lateral observations (Cronbach et al., 1972; Brennan, 2001) [19,20]. In view of this, the researchers should consider how to find a measurement program with a high reliability under the budget restriction in the evaluation of college teachers' teaching level.

Last but not least, the method of estimating the optimal sample size under budget constraints is explored. These methods are the discrete optimization methods, the Cauchy–Schwartz inequality method, and the Lagrange multiplier method, etc. Woodward and Joe (1973) [21] deduced the equation (maximize the reliability of the measurement under the budget limit) to estimate the sample size using the discrete optimization method. However, the discrete optimization method cannot be extended to more than three sides or above of the cross design or nesting design. Some scholars have improved these shortcomings (i.e., the Cauchy–Schwartz inequality method proposed by Sanders et al. (1989) [22], and the Lagrange multiplier method proposed by Macrolides and Goldstein (1990) [23]. Macrolides (1993) [24] generalizes the one-side Lagrange multiplier to the multi-lateral and multi-context in order to solve the problem of estimating the optimal sample size in multi-side design under budget constraints further. Moreover, Macrolides (1997) [25] conducts a "divide and conquer" strategy for the Lagrange multiplier method according to different generalizability designs. Macrolides (1997) [25] aims to simplify crossover designs instead, extending the focus of the research to more complicated nests or mixed designs (some generalizability designs are multifaceted mixed designs including both cross and nested designs) showing deficiencies in the discussion on the method of estimating the optimal sample size under budget constraints. The research of Li and Ou (2020) [26] shows that the performance of the Lagrange multiplier method is better than that of the

Cauchy–Schwartz inequality method. Therefore, this study only discusses the Lagrange multiplier method.

Meyer et al. (2014) [3] applied the Lagrange multiplier (estimating the optimal sample size of different designs in generalizability theory under budget constraints proposed by Macrolides et al. (1993, 1997) [24,25] to actual measurement research through an empirical study of the teaching level evaluation of college teachers. Meyer and other people list the application of the Lagrange multiplier in the simple crossover design and present the application of the Lagrange multiplier in the nested design, showing that the Lagrange multiplier has a wide applicability in the generalizability theory. However, there are still two questions in the study of Meyer et al. (2014) [3]. On the one hand, there is no unified formula for the Lagrange multiplier. Meyer et al. (2014) [3] still make a "divide and conquer" on different generalizability designs, and simply apply the basic principle of the Lagrange multiplier to different general designs, respectively, thereby reducing the optimal sample size of different generalizability designs. However, they do not put forward a relatively unified formula of the Lagrange multiplier, generalize, or summarize the basic principle of the Lagrange multiplier, showing deficiencies. On the other hand, they do not apply the Lagrange multiplier to more complicated generalizability designs. Although, the generalizability design of Meyer et al. (2014) [3] can apply the Lagrange multiplier to the nested design in generalizability theory. However, they cannot apply the Lagrange multiplier to some more complicated designs (i.e., the mixed design and multivariate conceptual design and etc.), which is deficient (Macrolides and Goldstein, 1991, 1992; Macrolides, 1994, 1995) [27–30].

Education is sustainable. If we want to maintain the sustainable development of higher education, we must have strong teachers and attach importance to college teachers' teaching level. In this view, we should strengthen the supervision of teachers' teaching behavior for the reason that the evaluation is very important, but it needs to be scientific and reasonable.

Budget and cost are the problems that cannot be neglected in the research of measurement. The cost will be higher when the number of students evaluating teachers is larger. Thus, it is impossible to expect to endlessly increase students to evaluate teachers' learning level. The size of a sample would be subject to budget constraints. Therefore, it is important to examine how to effectively determine the size of a sample considering the budget constraints. How many students and items are optimal for teaching level evaluation of college teachers? Possibly, we can find the answer from the generalizability theory and Lagrange multiplier method.

Generalization theory is widely used in various psychological evaluation practices (Clayson et al., 2021; Li, 2019; Vispoel et al., 2020) [31–34]. Generalizability coefficients can be improved by increasing sample sizes (Zhang and Lin, 2016) [35]. However, the size of a sample would be subject to budget constraints. When there is a budget constraint, the generalization theory needs to consider how to design a measurement program with relatively high reliability and feasibility, which requires the optimal sample size to be estimated by some means. The Lagrange multiplier method is a more mature method for estimating the optimal sample size under the budget constraints in generalizability theory.

The purpose of this study is as follows: (1) based on generalizability theory combined with the budget constraints, we unified the Lagrange multiplier formula and derived the optimal sample size estimators for different generalizability designs; (2) according to the optimal sample size estimators for different generalizability designs, we estimate the optimal number of students and the optimal number of items for the actual teaching level evaluation of college teachers, hoping to provide references for research similar to the evaluation of the teaching level of college teachers.

## 2. Lagrange Multiplier

### 2.1. The Unified Formula of Lagrange Multiplier

The Lagrange multiplier is a method of finding the extreme of a multivariate function whose variables are limited by one or more conditions. It can solve the problem of the

optimization with equality constraints by introducing the Lagrange multiplier (Wang and Wu, 1999) [36]. When using the Lagrange multiplier to solve a problem, the unified formula of the Lagrange function is formulated as follows:

$$L(x, y, \lambda) = f(x, y) - \lambda g(x, y) \tag{1}$$

where $L(x, y, \lambda)$ is the Lagrange function, x and y are unknown parameters of the function, $\lambda$ is a new unknown scalar. Equation (1) can be interpreted as solving the extremism of the function $f(x, y)$ under the constraint of the function $g(x, y)$ by introducing a new unknown scalar $\lambda$ (Zheng and Gao, 2018) [37].

According to Equation (1), if the teaching level evaluation of college teachers involves evaluation students-s, evaluation items-i, evaluation dimensions-v, and evaluation occasion (o, number of times), the Lagrange function can be expressed as:

$$L(n_s, n_i, n_v, n_o, \lambda) = \sigma^2(\delta) - \lambda(c n_s n_i n_v n_o - B) \tag{2}$$

where $n_s$ represents the number of evaluation students, $n_i$ represents the number of evaluation items, $n_v$ represents the number of the dimension of the evaluation items, $n_o$ represents the number of evaluation times, $\sigma^2(\delta)$ represents the relative error variance which corresponds to the error variance in CTT (Brennan, 1983, 2001; Qi et al., 2002) [14,19,38], $\lambda$ represents the new unknown scalar, c represents the cost of a single question, B represents the budget for completing an evaluation. Equation (2) set the function $\sigma^2(\delta)$ and the additional condition $(c n_s n_i n_v n_o - B) \leq 0$, introducing the unknown parameter $\lambda$; thereby finding the extreme value of the function $\sigma^2(\delta)$ under the restriction of additional conditions. $c n_s n_i n_v n_o - B \leq 0$ indicates the actual cost is less than or equal to the budget. When $c n_s n_i n_v n_o - B = 0$, the maximum value of $n_s n_i n_v n_o$ is obtained since c and B are fixed values. Therefore, Equation (2) adopts the extreme value of the relative error variance $\sigma^2(\delta)$ by introducing the Lagrange multiplier $\lambda$ as the actual cost is not greater than the budget.

According to Equation (2), the optimal sample size of all three designs can be estimated.

*2.2. The Lagrange Multiplier of the (s:t) × i Design*

The mean error variance and relative error variance of the (*s:t*) × *i* design are:

$$\sigma_{\overline{X}}^2 = \frac{\sigma_t^2}{n_t} + \frac{\sigma_i^2}{n_i} + \frac{\sigma_{s:t}^2}{n_s} + \frac{\sigma_{ti}^2}{n_i} + \frac{\sigma_{si:t,e}^2}{n_s n_i} \tag{3}$$

$$\sigma^2(\delta) = \frac{\sigma_{s:t}^2}{n_s} + \frac{\sigma_{ti}^2}{n_i} + \frac{\sigma_{si:t,e}^2}{n_s n_i} \tag{4}$$

In Equation (3), $\sigma_{\overline{X}}^2$ represents the mean error variance; $\sigma_t^2, \sigma_i^2, \sigma_{s:t}^2, \sigma_{ti}^2, \sigma_{si:t,e}^2$ are the variance components of the teachers, the variance component of the items, the variance component of the students nested in the teachers, the variance component of the teacher–item interaction, and the variance component of the students crossover the items, but nested in the teachers with unmeasured and random sources of variation, respectively. In this paper, we used the symbol 'e' to denote unmeasured and random sources of variation (Shavelson and Webb, 1991) [39]. The $n_t$, $n_i$ and $n_s$ represent the sample size of the teachers, the sample size of the items, and the sample size of the students, respectively. In Formula (4), $\sigma_\delta^2$ is the relative error variance, and the meanings of other representation symbols are the same as in Equation (3).

The optimal number of students and items can be obtained by conducting $\min F(n_s, n_i, \lambda) = \sigma^2(\delta) - \lambda(c n_s n_i - B)$ using the Lagrange multiplier unified formula of Equation (2) with Equations (3) and (4). Results are showed:

$$n_s = \sqrt{\frac{\sigma_{s:t}^2}{\sigma_{ti}^2} \frac{B}{c}} \tag{5}$$

$$n_i = \sqrt{\frac{\sigma_{ti}^2}{\sigma_{s:t}^2} \frac{B}{c}} \tag{6}$$

In Equation (5), $n_s$ represents the optimal number of students, $\sigma_{s:t}^2$ represents the estimated variance component of the students nested in the teachers, $\sigma_{ti}^2$ represents the estimated variance component of the teacher–item interaction, c represents the cost of a single item, $B$ represents the budget of completing an evaluation. In Equation (6), $n_i$ represents the optimal number of items, and the meanings of other representation symbols are the same as in Equation (5).

### 2.3. The Lagrange Multiplier of the (s:t) × (i:v) Design

The mean error variance and relative error variance of the (*s:t*) × (*i:v*) design are:

$$\sigma_{\overline{X}}^2 = \frac{\sigma_t^2}{n_t} + \frac{\sigma_v^2}{n_v} + \frac{\sigma_{i:v}^2}{n_i n_v} + \frac{\sigma_{s:t}^2}{n_s} + \frac{\sigma_{sv:t}^2}{n_s n_v} + \frac{\sigma_{tv}^2}{n_v} + \frac{\sigma_{ti:v}^2}{n_i n_v} + \frac{\sigma_{si:tv,e}^2}{n_s n_i n_v} \tag{7}$$

$$\sigma^2(\delta) = \frac{\sigma_{s:t}^2}{n_s} + \frac{\sigma_{sv:t}^2}{n_s n_v} + \frac{\sigma_{tv}^2}{n_v} + \frac{\sigma_{ti:v}^2}{n_i n_v} + \frac{\sigma_{si:tv,e}^2}{n_s n_i n_v} \tag{8}$$

In Equation (7), $\sigma_{\overline{X}}^2$ represents the mean error variance; $\sigma_t^2$, $\sigma_v^2$, $\sigma_{i:v}^2$, $\sigma_{s:t}^2$, $\sigma_{sv:t}^2$, $\sigma_{tv}^2$, $\sigma_{ti:v}^2$, $\sigma_{si:tv,e}^2$ are the variance components of the teachers, the variance component of the dimensions, the variance component of the items nested in the dimensions, the variance component of the students nested in the teachers, the variance component of the students crossover the dimensions, but nested in the teachers, the variance component of the teacher–dimension interaction, the variance component of the teachers crossover the items, but nested in the dimensions, and the variance component of the students crossover the items, but nested in the teachers and dimensions with unmeasured and random sources of variation, respectively. The $n_t$, $n_i$, $n_s$ and $n_v$ represent the sample size of the teachers, the sample size of the items, the sample size of the students, and the sample size of the dimensions, respectively. In Equation (8), $\sigma_\delta^2$ represents the relative error variance, and the meaning of other representation symbols are the same as in Equation (7).

The optimal number of students and items of the (*s:t*) × (*i:v*) design can be obtained by conducting $\min F(n_s, n_i, n_v, \lambda) = \sigma^2(\delta) - \lambda(c n_s n_i n_v - B)$ using the Lagrange multiplier formula of Equation (2) with Equations (7) and (8). Results are showed:

$$n_s = \sqrt{\frac{(n_v \sigma_{s:t}^2 + \sigma_{sv:t}^2)}{n_v \sigma_{ti:v}^2} \frac{B}{c}} \tag{9}$$

$$n_i = \sqrt{\frac{\sigma_{ti:v}^2}{n_v (n_v \sigma_{s:t}^2 + \sigma_{sv:t}^2)} \frac{B}{c}} \tag{10}$$

In Equation (9), $n_s$ represents the optimal number of students, $\sigma_{s:t}^2$ represents the estimated variance component of the students nested in the teachers, $\sigma_{sv:t}^2$ represents the estimated variance component of the students crossover the dimensions, but nested in the teachers, $\sigma_{ti:v}^2$ represents the estimated variance component of the teachers crossover the items, nested in the dimensions, $n_v$ represents the sample size of the dimensions, c represents the cost of a single item, $B$ represents the budget for completing an evaluation. In Equation (10), $n_i$ represents the optimal number of items, and the meanings of other representation symbols are the same as in Equation (9).

### 2.4. The Lagrange Multiplier of the (s:t) × (i:v) × o Design

The mean error variance and relative error variance of the (*s:t*) × (*i:v*) × o design are:

$$\sigma_{\overline{X}}^2 = \frac{\sigma_o^2}{n_o} + \frac{\sigma_t^2}{n_t} + \frac{\sigma_v^2}{n_v} + \frac{\sigma_{s:t}^2}{n_s} + \frac{\sigma_{i:v}^2}{n_i n_v} + \frac{\sigma_{ot}^2}{n_o} + \frac{\sigma_{os:t}^2}{n_s n_o} + \frac{\sigma_{ov}^2}{n_v n_o} + \frac{\sigma_{oi:v}^2}{n_i n_v n_o} + \frac{\sigma_{tv}^2}{n_v} + \frac{\sigma_{ti:v}^2}{n_i n_v} + \frac{\sigma_{sv:t}^2}{n_s n_v} + \frac{\sigma_{si:tv}^2}{n_s n_i n_v} + \frac{\sigma_{otv}^2}{n_v n_o} + \frac{\sigma_{oti:v}^2}{n_i n_v n_o} + \frac{\sigma_{osv:t}^2}{n_s n_v n_o} + \frac{\sigma_{osi:tv,e}^2}{n_s n_i n_v n_o} \tag{11}$$

$$\sigma_\delta^2 = \frac{\sigma_{s:t}^2}{n_s} + \frac{\sigma_{ot}^2}{n_o} + \frac{\sigma_{os:t}^2}{n_s n_o} + \frac{\sigma_{tv}^2}{n_v} + \frac{\sigma_{ti:v}^2}{n_i n_v} + \frac{\sigma_{sv:t}^2}{n_s n_v} + \frac{\sigma_{si:tv}^2}{n_s n_i n_v} + \frac{\sigma_{otv}^2}{n_v n_o} + \frac{\sigma_{oti:v}^2}{n_i n_v n_o} + \frac{\sigma_{osv:t}^2}{n_s n_v n_o} + \frac{\sigma_{osi:tv,e}^2}{n_s n_i n_v n_o} \qquad (12)$$

In Equation (11), $\sigma_{\overline{X}}^2$ means the mean error variance; $\sigma_o^2$, $\sigma_t^2$, $\sigma_v^2$, $\sigma_{s:t}^2$, $\sigma_{i:v}^2$, $\sigma_{ot}^2$, $\sigma_{os:t}^2$, $\sigma_{ov}^2$, $\sigma_{oi:v}^2$, $\sigma_{tv}^2$, $\sigma_{ti:v}^2$, $\sigma_{sv:t}^2$, $\sigma_{si:tv}^2$, $\sigma_{otv}^2$, $\sigma_{oti:v}^2$, $\sigma_{osv:t}^2$, $\sigma_{osi:tv,e}^2$ represent the variance component of the occasions; the variance component of the teachers; the variance component of the dimensions; the variance component of the students nested in the teachers; the variance component of the items nested in the dimensions; the variance component of the interaction between the teachers and occasions; the variance component of the interaction between the occasions and students, but nested in the teachers; the variance component of the interaction between the occasions and dimensions; the variance component of the interaction between the occasions and items, but nested in the dimensions; the variance component of the interaction between the teachers and dimensions; the variance component of the interaction between the teachers and items, but nested in the dimensions; the variance component of the students crossover the dimensions, but nested in the teachers; the variance component of the interaction between the students and the items, but nested in the teachers crossover the dimensions; the variance component of the interaction of occasion–teacher–dimension; the variance component of the interaction of occasion–teacher–item, but nested in the dimensions; the variance component of the interaction of occasion–student–dimension, but nested in the teachers crossover the dimensions with unmeasured and random sources of variation, respectively. The $n_t$, $n_i$, $n_s$, $n_v$ and $n_o$ are the sample size of teachers, the sample size of items, the sample size of students, the sample size of dimensions, and the sample size of occasions, respectively. In Equation (12), $\sigma_\delta^2$ represents the relative error variance, and the meanings of other representation symbols are the same as in Equation (11).

The optimal number of students and items can be obtained by conducting $\min F(n_s, n_i, n_v, n_o, \lambda) = \sigma^2(\delta) - \lambda(c n_s n_i n_v n_o - B)$ using the Lagrange multiplier formula of Equation (2) with Equations (11) and (12). Results are showed:

$$n_s = \sqrt{\frac{\left(n_v n_o \sigma_{s:t}^2 + n_v \sigma_{os:t}^2 + n_o \sigma_{sv:t}^2 + \sigma_{osv:t}^2\right)}{n_v n_o \left(n_o \sigma_{ti:v}^2 + \sigma_{oti:v}^2\right)} \frac{B}{c}} \qquad (13)$$

$$n_i = \sqrt{\frac{n_o \sigma_{ti:v}^2 + \sigma_{oti:v}^2}{n_v n_o \left(n_v n_o \sigma_{s:t}^2 + n_v \sigma_{os:t}^2 + n_o \sigma_{sv:t}^2 + \sigma_{osv:t}^2\right)} \frac{B}{c}} \qquad (14)$$

In Equation (13), $n_s$ represents the optimal number of students, $\sigma_{s:t}^2$ represents the estimated variance component of the students nested in the teacher, $\sigma_{os:t}^2$ represents the variance component of the situations crossover the students, but nested in the teachers, $\sigma_{sv:t}^2$ represents the estimated variance component of students crossover the dimensions, but nested in the teachers, $\sigma_{osv:t}^2$ represents the variance component of the occasion–student–dimension interaction, but nested in the teachers, $\sigma_{ti:v}^2$ represents the estimated variance component of the teachers crossover the items, but nested in the dimensions, $\sigma_{oti:v}^2$ represents the variance component of the occasion–teacher–item interaction, but nested in the dimensions, $n_v$ represents the sample size of the dimensions, $n_o$ represents the sample size of the occasions, c represents the unit cost, $B$ represents the budget to complete an evaluation. In Equation (14), $n_i$ represents the optimal number of the items, and the meanings of other representation symbols are the same as in Equation (13).

## 3. Methods

### 3.1. Data Collection

The Teaching Level Evaluation Questionnaire for College Teachers was used to evaluate the teaching level of college teachers. The main subjects participating in the questionnaire were trained, and the instructions were unified, and the objectives and points of attention were stated before the formal test.

The questionnaire includes three dimensions (teaching method, teaching content, and teaching effect). The teaching method is the general term of the ways and means used by teachers and students in the teaching process to achieve the common teaching goals and complete the common teaching tasks. The teaching content is the main information that is intentionally transmitted in the process of interaction between learning and teaching, generally including curriculum standards, textbooks, and courses. The teaching effect refers to the effect achieved by teachers through teaching behavior to guide students to acquire knowledge, such as better academic performance and greater progress made by students. Each dimension has 18 items, and there are 54 items in total on a 5-point scale (from 1 = Disagree at all to 5 = Agree very much). The Cronbach's $\alpha$ coefficient of the whole questionnaire is 0.88. The internal consistency coefficients of all dimensions and the questionnaire are 0.85, 0.80, 0.85, and 0.94. The correlation between the scores of all dimensions and the total score of the questionnaire was 0.62~0.78. We performed a series of confirmatory factor analyses (CFA) (Geiser, 2012) [40] to identify the dimensions of the scales using a pilot sample data ($n$ = 501) collected in 2020, prior to our formal research. Results indicated three dimensions (and each of these dimensions was examined using 18 items and averaged): teaching method (e.g., "the teacher is good at using multi-media, such as lantern slide, models, films, for teaching; the teacher summarizes and emphasizes the key points and difficult points clearly."), teaching content (e.g., "the teacher introduces us the present trend of the subject and the background of the learning content; the teacher links learning content to practical life."), and teaching effect (e.g., "through the study, I grasp the basic principles and theories of the curriculum; through the study, I learn how to solve problems by searching and using information resources."). The model fit was acceptable (Schermelleh-Engel et al., 2003) [41] and the validity of the questionnaire met the measurement requirements ($\chi^2/df$ = 3.283, CFI = 0.927, TLI = 0.918, RMSEA = 0.066, SRMR = 0.039).

Table 1 shows the number of teachers and students for different major type participating in the evaluation of the teaching level for college teachers.

**Table 1.** The number of students of the teacher for different major type in the evaluation.

| Teacher ID | 1 | 2 | 3 | 4 | 5 | 6 | 7 | 8 | 9 | 10 |
|---|---|---|---|---|---|---|---|---|---|---|
| Student number | 43 | 39 | 42 | 39 | 39 | 37 | 38 | 42 | 39 | 39 |
| Major type | A | S | A | E | E | A | A | S | E | S |

Note: A = liberal arts; S = science; E = engineering.

As shown in Table 1, there are 10 teachers who are evaluated and 397 students who participate in the evaluation. The types of students include liberal arts, science, and engineering.

### 3.2. Procedures

In total, we investigated 10 teachers (and 10 classes or 10 courses) from three colleges and their teaching performance. All of the 10 courses were mandatory, and had the same workload (one 45 min lesson a week). Data were collected during the class in 45 min using a paper/pencil version survey administered to all students in these classes, first at the end of the first semester (T1, before the final exam; fall semester) and then at the beginning of the second semester (T2, spring semester). Research staff was trained before they administered the survey. Student assent was obtained, and this study received approval documents from the targeted university's research ethics board (Institutional Review Board).

### 3.3. Generalizability Design

The generalizability design includes (*s:t*) × *i*, (*s:t*) × (*i:v*) and (*s:t*) × (*i:v*) × *o*, where *t* is the object of measurement, *i*, *s*, *v*, and *o* are facets of measurement. Among them, *t* denotes the evaluated teacher, *i* is the evaluated item, *s* denotes the evaluated student, *v* represents

the dimension of the evaluation item (dimension or veidoo), and *o* denotes the evaluated occasion (occasion).

### 3.4. Analysis Tools

Analyses were performed in the urGENOVA software (Brennan, 2001) [42]. Some programs were completed by writing the control card in the urGENOVA software. Some variance components were estimated on each side of the data in different designs, and then substituted the variance components into the unified formula of the Lagrange multiplier to calculate the optimal sample size for different designs under budget constraints.

### 3.5. Budgetary Costs

The budget constraints and measurement costs from the actual situation of the evaluation of teaching level of college teachers are set. C represents the single-item cost, which is the cost of completing an item in the evaluation questionnaire. The cost of completing an evaluation questionnaire (a total of 54 items) is showed in Table 2.

**Table 2.** The cost of completing an evaluation questionnaire (54 items in total).

| Details | Price | Cost of a Single Question C |
|---|---|---|
| Design cost of the evaluation item | CNY 4/share | |
| Printing cost of the evaluation item | CNY 0.8/share | c = CNY 0.2 |
| Cost of the analysis of evaluation data | CNY 6/share | |

The cost of a single item is CNY 0.2; the cost of completing an evaluation questionnaire (54 items in total) is CNY 10 ($54 \times 0.2 = 10.8$). This paper defines the budget needed to complete the evaluation based on the cost of completing an evaluation questionnaire. $B^* = B \times k$ represents the total budget for completing k times evaluation, where $B$ is the budget for completing one time evaluation. The budget cost of the evaluation can be set according to the actual needs. For example, there are a total of 397 evaluation questionnaires in the evaluation of teaching level of college teachers. If the evaluation is conducted twice ($k = 2$), the total budget will be $B^* = $ CNY 4287.6.

The $cn_s n_i \leq B$ is a budget constraint expression of two-sided design according to the definition of budgetary cost, where $n_s$ indicates the number of students participating in the evaluation (or the number of completed evaluation questionnaires), $n_i$ is the number of evaluation questionnaire questions, and $cn_s n_i \leq B$ indicates the cost of one time that must be less than or equal to the predetermined budget cost in the evaluation of college teachers' teaching level. Similarly, the budget constraint expression can be expressed as $cn_s n_i n_v \leq B$ if the definition of the budget cost is extended to a three-sided design, where $n_s$ indicates the number of students participating in the evaluation (or the number of completed questionnaires), $n_i$ is the number of items contained in each dimension, $n_v$ indicates the number of the dimension in each evaluation items. When more aspects are involved, the expression of the budget constraint can be expressed as $cn_s n_i n_v \cdots n_x \leq B$.

## 4. Results

### 4.1. The Optimal Sample Size Estimation for the (s:t) × i Design

The size of the test population is a non-negligible factor affecting the evaluation results (Hill et al., 2012) [8]. Most of the evaluations are based on sampling methods due to the practical limitations of the operation; thus, the size of the sample becomes an important factor influencing the effectiveness of the evaluation. It is necessary to consider the impact of this factor in the evaluation of the teaching level of college teachers. In general, the evaluation student-s is nested within the evaluated teacher-t and intersects with the evaluation item-i. Their relationship is (*s:t*) × *i*.

Results about the estimated variance component of the (*s:t*) × *i* design from the performance of the urGENOVA program are showed in Tables 3 and 4.

**Table 3.** ANOVA result of urGENOVA for (*s:t*) × *i*.

| Effect | df | T | SS | MS |
|---|---|---|---|---|
| *t* | 9 | 298,136.60505 | 816.58597 | 90.73177 |
| *s:t* | 387 | 304,396.20370 | 6259.59865 | 16.17467 |
| *i* | 53 | 300,299.11587 | 2979.09679 | 56.20937 |
| *ti* | 477 | 302,017.40826 | 901.70641 | 1.89037 |
| *si:t, e* | 20511 | 336,825.00000 | 28,547.99309 | 1.39184 |

**Table 4.** The Variance Components Estimated of the (*s:t*) × *i* Design.

| *t* | *s:t* | *i* | *ti* | *si:t, e* |
|---|---|---|---|---|
| 0.03455 | 0.27376 | 0.13682 | 0.01256 | 1.39184 |

The calculation is available as $n_s$ = 521.96957, $n_i$ = 23.94775 based on the variance components estimated by Equations (5) and (6) and Table 4. The results of $n_s$ and $n_i$ are rounded to 522 and 24; the total cost is 2496 and the relative error variance $\sigma^2(\delta)$ is 0.00116 when $n_s$ = 522, $n_i$ = 24, $E\rho^2$ = 0.96752.

When the number of evaluation student is 52 for each teacher and the number of the evaluation items is 8 items for each dimension in the (*s:t*) × *i* design, the reliability of the evaluation is the greatest.

*4.2. The Optimal Sample Size Estimation for the (s:t) × (i:v) Design*

The reliability of the evaluation result is not only influenced by the size of the test group, but also by the setting of the evaluation items. When assessing the teaching level of college teachers, it is also possible to examine the influence of the dimension of the evaluation items on evaluation results so as to make the analysis results more comprehensive. The relationship among the evaluation students, evaluation items, and the dimension of the evaluation items is: the students-s are nested in the evaluated teachers-t, while the evaluative items-i are nested in the dimension-v of the evaluative items, and the relationship between them is (*s:t*) × (*i:v*).

The results of the estimated variance component of the (*s:t*) × (*i:v*) design from the performance of the urGENOVA program are shown in Tables 5 and 6.

**Table 5.** ANOVA result of urGENOVA for (*s:t*) × (*i:v*).

| Effect | df | T | SS | MS |
|---|---|---|---|---|
| *t* | 9 | 298,136.60505 | 816.58597 | 90.73177 |
| *s:t* | 387 | 304,396.20370 | 6259.59865 | 16.17467 |
| *v* | 2 | 297,396.69728 | 76.67820 | 38.33910 |
| *i:v* | 51 | 300,299.11587 | 2902.41859 | 56.91017 |
| *tv* | 18 | 298,300.29952 | 87.01626 | 4.83424 |
| *ti:v* | 459 | 302,017.40826 | 814.69015 | 1.77492 |
| *sv:t* | 774 | 306,288.76254 | 1728.86437 | 2.23367 |
| *si:tv, e* | 19,737 | 336,825.00000 | 26,819.12872 | 1.35882 |

**Table 6.** The Variance Components Estimated of the (*s:t*) × (*i:v*) Design.

| *t* | *s:t* | *v* | *i:v* | *tv* | *ti:v* | *sv:t* | *si:tv, e* |
|---|---|---|---|---|---|---|---|
| 0.03349 | 0.25688 | −0.00311 | 0.13888 | 0.00314 | 0.01048 | 0.04989 | 1.35882 |

The calculation is available as $n_s$ = 571.16419, $n_i$ = 7.29504 based on the variance component estimated by Equations (9) and (10) and Table 6 (the number of items in each dimension). The results of $n_s$ and $n_i$ are rounded to 571 and 7. In the (*s:t*) × (*i:v*) design; the

total cost is 2398.2 and the relative error variance $\sigma^2(\delta)$ is 0.00109 when $n_s = 522$, $n_i = 24$, $E\rho^2 = 0.96848$.

When the number of evaluation student is 57 for each teacher and the number of the evaluation items is 7 items for each dimension in the $(s{:}t) \times (i{:}v)$ design, the reliability of the evaluation is the greatest.

*4.3. The Optimal Sample Size Estimation for the $(s{:}t) \times (i{:}v) \times o$ Design*

On the evaluation of college teachers' teaching level, it is necessary to evaluate the influence of the occasion (time or the number of times) in addition to the size of the group of participants and the setting of evaluation items. The measurement results on different occasions (re-tested) can make the evaluation result more persuasive to a certain extent. However, it also means the cost will increase. In the assessment of college teachers' teaching level, the evaluation student-s is nested in the evaluated teacher t, and the evaluation item-i is nested in the dimension-v of the evaluation items, and there is a cross relationship between them. If the evaluation occasion (time or the number of times) is included in the generalizability design, the two nests intersect with the evaluation occasion (time or the number of times) o, namely $(s{:}t) \times (i{:}v) \times o$.

Those results of the estimated variance components of the $(s{:}t) \times (i{:}v) \times o$ design from the performance of the urGENOVA program are shown in Tables 7 and 8.

**Table 7.** ANOVA result of urGENOVA for $(s{:}t) \times (i{:}v) \times o$.

| Effect | df | T | SS | MS |
|---|---|---|---|---|
| o | 1 | 603,244.46427 | 30.68404 | 30.68404 |
| t | 9 | 603,950.41673 | 736.63650 | 81.84850 |
| s:t | 387 | 609,305.87963 | 5355.46290 | 13.83841 |
| v | 2 | 603,391.54635 | 177.76613 | 88.88306 |
| i:v | 51 | 609,157.78212 | 5766.23576 | 113.06345 |
| ot | 9 | 604,541.19110 | 560.09034 | 62.23226 |
| os:t | 387 | 615,196.01852 | 5299.36451 | 13.69345 |
| ov | 2 | 603,423.19852 | 0.96812 | 0.48406 |
| oi:v | 51 | 609,374.12343 | 184.68914 | 3.62136 |
| tv | 18 | 604,233.40471 | 105.22185 | 5.84566 |
| ti:v | 459 | 610,968.65628 | 969.01581 | 2.11115 |
| sv:t | 774 | 611,401.23105 | 1812.36344 | 2.34155 |
| si:tv | 19,737 | 647,921.50000 | 29,785.01738 | 1.50910 |
| otv | 18 | 604,872.81689 | 47.66969 | 2.64832 |
| oti:v | 459 | 612,708.89276 | 916.13515 | 1.99594 |
| osv:t | 774 | 619,097.49034 | 1757.48259 | 2.27065 |
| osi:tv, e | 19,737 | 686,117.00000 | 29,398.41641 | 1.48951 |

**Table 8.** The Variance Components Estimated of the $(s{:}t) \times (i{:}v) \times o$ Design.

| o | t | v | s:t | i:v | ot | os:t | ov | oi:v |
|---|---|---|---|---|---|---|---|---|
| −0.00136 | 0.08376 | −0.00173 | 0.00065 | 0.13769 | 0.02247 | 0.2139 | −0.00054 | 0.00409 |

| tv | ti:v | sv:t | si:tv | otv | oti:v | osv:t | osi:tv, e |
|---|---|---|---|---|---|---|---|
| 0.00218 | 0.00120 | 0.00146 | 0.00979 | −0.00018 | 0.01276 | 0.04454 | 1.48951 |

The calculation is available as $n_s = 305.26026$, $n_i = 6.80250$ based on the variance components estimated by Equations (13) and (14) and Table 8. The results of $n_s$ and $n_i$ are rounded to 306 and 7. In the $(s{:}t) \times (i{:}v) \times o$ design, the total cost is 1285.2 and the relative error variance $\sigma^2(\delta)$ are 0.001264 when $n_s = 306$, $n_i = 7$, $E\rho^2 = 0.86888$.

When the number of evaluation students is 31 for each teacher and the number of the evaluation items is 7 items for each dimension, $n_v = 3$, $n_o = 2$, the reliability of the evaluation is the greatest.

## 5. Discussion

### 5.1. The Influencing Factors of the Evaluation

The generalizability theory replaces the notion of reliability in the CTT with the concept of dependability, which refers to extend the scores of the participants in one measurement (such as psychological test, behavioral observation, questionnaire survey, etc.) to the accuracy of the generalizability of the scores of participants on all possible conditions accepted by the test participants at the same level. The concept of dependability in generalizability theory can be expressed as a generalizability coefficient $E\rho^2$ (Brennan, 2001) [19]:

$$E\rho^2 = \frac{\sigma^2(t)}{\sigma^2(t) + \sigma^2(\delta)} \tag{15}$$

where $E\rho^2$ represents the generalizability coefficient, $\sigma^2(t)$ is the variance of the global fractional (means the teacher's global fractional variance in this case), and $\sigma^2(\delta)$ represents the relative error variance. In Equation (15), the generalizability coefficient $E\rho^2$ can be defined as the ratio of the sum of the global fractional variance and relative error variance.

The global fractional variance and relative error variance of all three designs can be obtained according to the results of Tables 3–6 and the above analysis, and the optimal generalizability coefficients $E\rho^2$ and $\Delta E\rho^2$ of all three designs can be calculated according to Equation (15), which is shown in Table 9.

**Table 9.** The Optimal generalizability coefficients and Their Changes.

| Design | Global Fractional Variance $\sigma^2(t)$ | Relative Error Variance $\sigma^2(\delta)$ | Optimal Generalizability Coefficient $E\rho^2$ | $\Delta E\rho^2$ |
|---|---|---|---|---|
| $(s{:}t) \times i$ | 0.03455 | 0.00116 | 0.96752 | |
| $(s{:}t) \times (i{:}v)$ | 0.03349 | 0.00109 | 0.96848 | $-0.00096$ |
| $(s{:}t) \times (i{:}v) \times o$ | 0.08376 | 0.01264 | 0.86888 | 0.09960 |

From Table 9, the complexity of all three designs is continuously increasing, showing a progressive relationship. The next design is formed by adding a factor to the previous design, which means that the design of $(s{:}t) \times (i{:}v) \times o$ is based on the design of $(s{:}t) \times i$ with the effect of v added. The $(s{:}t) \times (i{:}v) \times o$ design is formed after adding the o factor to the $(s{:}t) \times (i{:}v)$ design. The advantage is that it can intuitively dig out the influence of each factor on the generalizability coefficient, thus it can explore more hidden facets and lay the foundation for discovering the main influencing factors of the evaluation of college teachers' teaching level.

From Table 9, these facets gradually emerge from the hidden facets with the facets added one by one, which results in the "fixed" variance in the global fractional as being gradually "liberated" (Brennan, 2001) [19]. This also shows that these "liberated" facets may have a certain impact on the evaluation of the teaching level of college teachers, which is consistent with the research of Meyer et al. (2014) [3]. The $(s{:}t) \times (i{:}v) \times o$ design has a relatively large change in the generalizability coefficient ($\Delta E\rho^2$) compared to the $(s{:}t) \times (i{:}v)$ design, reaching 0.09960 (9.960%), showing that o (occasion) is an important factor in evaluating the teaching level of college teachers. However, the $(s{:}t) \times (i{:}v)$ design has a relatively smaller change in the generalizability coefficient ($\Delta E\rho^2$) than the $(s{:}t) \times i$ design, only $-0.00096$ (0.096%), indicating that the v (Dimensions) can only be considered as a general factor.

### 5.2. The Optimal Sample Size Estimation for Different Designs

The estimated optimal sample size for all three designs can be obtained by sorting out the data in the above results, as shown in Table 10.

**Table 10.** The Estimated Optimal Sample Size for the Three Designs.

| Design | $n_s$ | $n_i$ | $n'_s$ | $n'_i$ | $n_s n_i$ | $n'_s n'_i$ |
|:---:|:---:|:---:|:---:|:---:|:---:|:---:|
| $(s{:}t) \times i$ | 520 | 24 | 52 | 8 | 12,480 | 416 |
| $(s{:}t) \times (i{:}v)$ | 571 | 21 | 57 | 7 | 11,991 | 399 |
| $(s{:}t) \times (i{:}v) \times o$ | 306 | 21 | 31 | 7 | 6426 | 217 |

In Table 10, $n_s$ and $n_i$ indicate the total number of the optimal students and items needed to evaluate the 10 teachers. In Table 10, $n'_s$ is the average number of the optimal students needed to evaluate per teacher, $n'_i$ indicates the average number of optimal items contained in the dimensions.

In Table 10, the optimal number of students in all three designs is 520, 571, and 306 while the optimal number of items for all three designs is 24, 21, and 21. The average number of optimal students in all three designs is 8, 7, and 7. Comparing the $(s{:}t) \times (i{:}v)$ and $(s{:}t) \times (i{:}v) \times o$ designs in Table 11, the former's average number of optimal students and items are 57 and 7, and the latter's average number of optimal students and items are 31 and 7 with a great difference between them. In addition, the product of $(s{:}t) \times (i{:}v)$ and $(s{:}t) \times (i{:}v) \times o$ is relatively big under the $n_s n_i n'_s n'_i$ columns of data in Table 11. The former is 11,991 and 399, and the latter is 6426 and 217, indicating that the gap between them is relatively big.

**Table 11.** The Budget Costs Corresponding to the Optimal Sample Sizes of all Three Designs.

| Design | Optimal Generalizability Coefficient $E\rho^2$ | $n_s$ | $n_i$ | Actual Budget Cost $B$ | $\triangle B$ |
|:---:|:---:|:---:|:---:|:---:|:---:|
| $(s{:}t) \times i$ | 0.96752 | 520 | 24 | 2496 | $-4$ |
| $(s{:}t) \times (i{:}v)$ | 0.96848 | 571 | 21 | 2398.2 | $-101.8$ |
| $(s{:}t) \times (i{:}v) \times o$ | 0.86888 | 306 | 21 | 1285.2 | $-1214$ |

Compared to the $(s{:}t) \times (i{:}v)$ design, the $(s{:}t) \times (i{:}v) \times o$ design adds a factor o, which will decompose the effect of the important factor "occasion". From the analysis, this occasion factor is very important, which is consistent with the research of Iqbal et al. (2016) [43]. According to this, it will have a great impact on the estimation of the optimal sample size from the above and we can see that the gap between the number of optimal students and items of the two designs is big. So, the $(s{:}t) \times (i{:}v) \times o$ design is superior.

The $(s{:}t) \times (i{:}v) \times o$ design under the budget constraint has a higher efficiency for estimating the optimal sample size compared with other designs.

*5.3. The Optimal Generalizability Design under Budget Constraints*

The budget costs corresponding to the optimal sample size of all three designs can be further summarized. Results are shown in Table 11.

In Table 11, $\Delta B$ is actual budget cost $B$ minus 2500, where the actual budget cost $B = 0.2\, n_s n_i$ is the actual corresponding budget cost generated after rounding off the optimal number of students and items, and CNY 2500 is the theoretical budget cost set at the beginning.

The cost budgets of the $(s{:}t) \times (i{:}v) \times o$ design are least under the $B$ and $\triangle B$ of all three designs in Table 11, which is CNY 1285.2 ($-1214$) and will greatly increase the effectiveness of the designs. This means that the $(s{:}t) \times (i{:}v) \times o$ designs show strong applicability under certain budgetary constraints and should be given priority.

From the generalizability coefficient of all three designs in Table 11, the $(s{:}t) \times (i{:}v)$ design has the largest generalizability coefficient (0.96848), followed by the $(s{:}t) \times i$ design (0.96752), and the $(s{:}t) \times (i{:}v) \times o$ design (0.86888). Although the generalizability coefficient of the $(s{:}t) \times (i{:}v) \times o$ design is the lowest in all three design, it involves the largest number of measurement facets. In addition, the generalizability coefficient of all three designs is

already very high as far as its size of the generalizability coefficient and the smallest one has reached 0.86888, which completely guarantees the reliability of the evaluation.

The (*s:t*) × (*i:v*) × *o* design analyses the results more comprehensively and is in line with the actual situation of the assessment of teaching level of college teachers. In addition, it includes the evaluator students, the evaluation items, and the dimension of the evaluation items and the measurement of evaluation on different occasions (time or the number of times). In fact, there are differences in the evaluation of teachers by different students on different occasions (time or the number of times), which indicates that the evaluation occasions (time or the number of times) have a considerable influence on the evaluation. Especially when students are under the pressure of the exam, they have to make high scores on the teacher's evaluation (Vaillancourt, 2013; Iqbal et al., 2016) [43,44].

Considering the above three aspects of budget cost, generalizability coefficient, and actual situation, the (*s:t*) × (*i:v*) × *o* design is the optimal generalizability design under the budget constraint of the college teacher's teaching level evaluation in the generalizability theory.

Under budget constraints, the number of students can be reduced, even to about 30. If the evaluation occasion or time is appropriate, the questionnaire evaluation can also ensure considerable reliability. So, this gives us two inspirations: on the one hand, it is not necessary to implement large-scale questionnaire evaluation. A considerable number of students can also ensure the reliability of the evaluation, which saves the human, material, and financial resources of colleges and universities; on the other hand, the evaluation occasion or time can be adjusted according to the actual situation, and the number of students and items in the evaluation can be appropriately reduced to the optimal sample size of students as 31 for each teacher and the optimal sample size of items as 7 for each dimension. It is not necessary to carry out large-scale questionnaire evaluations to obtain enough students and items, which is usually seen as more insurance. In fact, if the occasion or time is appropriate, a certain number of students and items will be enough.

### 5.4. Limitations

For this paper, there are the following limitations. Firstly, the teacher sample is relatively small, so we can increase the collection of teacher samples in the future to strengthen the representativeness of teacher samples. Secondly, missing data are not considered in this study. This study has collected data twice, but the collected data are all full data, and there are no gaps. In the future, we can consider how to deal with the missing data. Last but not least, more influencing factors can also be considered. This study only considers four influencing factors: students, items, dimensions, and occasions. In fact, other influencing factors, such as courses and majors, can also be considered in future research.

### 6. Conclusions

(1) The unified formula of the Lagrange multiplier shows a strong universality, which can be applied to various designs of generalizability theory under the budget constraint. Using the unified formula of the Lagrange multiplier and the estimated variance component of different generalizability designs, the optimal sample size for minimizing the relative error variance to the most and maximizing the generalizability coefficient under different generalizability theory designs can be deduced under budget constraints.

(2) Occasion is a very important factor influencing the evaluation of college teachers' teaching level. Compared with the other two factors, students and dimensions, the variation of the generalizability coefficient in the occasion is the largest. Therefore, this factor can maximize the ability to "liberate" more "fixed" variance effectively in global fractions with a very important influence on the evaluation of college teachers' teaching level.

(3) The (*s:t*) × (*i:v*) × *o* design has a higher efficiency in estimating the optimal sample size for the teaching level of college teachers. It can decompose the effect of the important influencing factors of the evaluation situation (time or the number of times), and it

        is also relatively small in the product of the estimated number of optimal students and items.

(4)    The $(s{:}t) \times (i{:}v) \times o$ design is the optimal generalizability design of the teaching level evaluation for college teachers under budget constraints in generalizability theory. Comparing the performance of the optimal sample size estimation for the $(s{:}t) \times I$, $(s{:}t) \times (i{:}v)$, and $(s{:}t) \times (i{:}v) \times o$ design comprehensively, the $(s{:}t) \times (i{:}v) \times o$ design is the optimal design under the three aspects of the budget, namely cost, generalizability coefficient, and actual situation.

(5)    Under budget constraints of teaching level evaluation for college teachers in generalizability theory, the optimal sample size of students is 31 for each teacher and the optimal sample size of items is 7 for each dimension.

**Funding:** This research was supported in part by Grant No. 2021A1515012516 from the Natural Science Foundation of Guangdong Province.

**Institutional Review Board Statement:** Not applicable.

**Informed Consent Statement:** Informed consent was obtained from all subjects involved in the study.

**Data Availability Statement:** The data are not publicly available due to privacy restrictions.

**Conflicts of Interest:** The author declares no conflict of interest.

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
