# Peer review of "How Many Students and Items Are Optimal for Teaching Level Evaluation of College Teachers? Evidence from Generalizability Theory and Lagrange Multiplier"

_sustainability, doi:10.3390/su15010002_

Round 1
Reviewer 1 Report
This study was conducted to figure out the optimal sample size of students and of question items in the evaluation of the teaching level of college teachers by using the LaGrange multiplier. The results showed that the formula of LaGrange multiplier could yield a robust result across different research designs under budget constraint. Additionally, the variable of occasion was found influential in teaching level evaluation for college teachers. Accordingly, the(s:t)×(i:v)×o design was efficient in estimating the optimal sample size of teaching level evaluation under budget constraints. Finally, the optimal sample size of students was found to be 31 students and the optimal sample size of items was found to be 7 questions. This study has its own merits, pointing out a practical problem in the evaluation of teaching level of college teachers and significantly contributing to the field of teaching level evaluation research. Although this study merits its own investigation, there are still some issues which need addressing. I will list my point-by-point comments as follows.
1. Since this paper is an academic study, it is suggested that the following sentences should be written in complete clauses.
a. First, build different generalizability designs.
b. Second, consider the budget and cost of the evaluation.
c. Third, explore the method of estimating the optimal sample size under budget constraints.
d. Set the budget constraints and measurement costs from the actual situation of the evaluation of teaching level of college teachers.
2. The following sentences are too long for readers to easily follow because they convey too many meanings. I suggest these sentences should be separated into several sentences.
a. Examining the influence of these factors on the teaching level of college teachers can help to accurately evaluate whether or not a teacher's teaching effect is good, and the level is high, and it also helps to understand the overall teaching level of a teacher from multiple aspects better and compare the differences between the teaching levels of different teachers fully.
b. It is necessary to pay attention to these influencing factors on the objects of measurement and constructing different generalizability designs to reflect the relationship among them in the analysis process using generalizability theory, otherwise they may become hidden facets thereby exaggerating the generalized coefficient of the evaluation (Brennan, 2001a).
c. In Table 10, ns and ni indicate the total number of the optimal students and projects needed to evaluate the 10 teachers, n's is the average number of the optimal students needed to evaluate per teacher, n'i indicates the average number of optimal topics contained in the dimensions.
3. The author mentioned that the Cauchy-Shiya inequality method and the LaGrange multiplier method were proposed to improve the discrete optimization method. The author should explain why the LaGrange multiplier method was used in stead of the Cauchy-Shiya inequality method in this study.
4. A research question should be proposed based on the research purpose.
5. This word “research” is generally uncountable. If the word becomes countable, that will mean research in different fields/areas. Please make sure the wording is correct in the study.
6. Because it is the main research instrument in this study, more descriptions of the Teaching Level Evaluation Questionnaire for College Teachers should be provided.
7. The author mentioned that there were three dimensions in the questionnaire. Each dimension consisted of 18 questions, and there were 54 titles in total. It is suggested that the author should provide two or three sample questions in each item. In addition, what do those titles refer to?
8. It’s better to report the reliability coefficient of the overall questionnaire first and then the reliability coefficients of the three dimensions.
9. Please explain the following sentence more because it looks confusing. “The average correlation between the scores of all dimensions and the total score of the questionnaire was 0.82.”
10. The author mentioned that Table 1 shows the distribution of the number of students and types of students participating in the evaluation of the teaching level of college teachers. I cannot tell any type of students in Table 1.
11. How were the 10 teachers selected to participate in this study? What sampling method was adopted in this study? Were they at the same college?
12. It is suggested that the author should define the four factors (i.e. students, topics, dimensions and occasions) in the section of introduction or methods.
13. Pedagogical implications should be provided based on the findings obtained in this study.
14. The reference list should be updated.
Author Response
This study was conducted to figure out the optimal sample size of students and of question items in the evaluation of the teaching level of college teachers by using the LaGrange multiplier. The results showed that the formula of LaGrange multiplier could yield a robust result across different research designs under budget constraint. Additionally, the variable of occasion was found influential in teaching level evaluation for college teachers. Accordingly, the(s:t)×(i:v)×o design was efficient in estimating the optimal sample size of teaching level evaluation under budget constraints. Finally, the optimal sample size of students was found to be 31 students and the optimal sample size of items was found to be 7 questions. This study has its own merits, pointing out a practical problem in the evaluation of teaching level of college teachers and significantly contributing to the field of teaching level evaluation research. Although this study merits its own investigation, there are still some issues which need addressing. I will list my point-by-point comments as follows.
- Since this paper is an academic study, it is suggested that the following sentences should be written in complete clauses.
- First, build different generalizability designs.
Answer: Thanks for your comments! We have revised as:
First of all, different generalizability designs are built. All changes are marked with red lines.
- Second, consider the budget and cost of the evaluation.
Answer: Thanks for your comments! We have revised as:
Secondly, the budget and cost of the evaluation are considered.
- Third, explore the method of estimating the optimal sample size under budget constraints.
Answer: Thanks for your comments! We have revised as:
Finally, the method of estimating the optimal sample size under budget constraints is explored.
- Set the budget constraints and measurement costs from the actual situation of the evaluation of teaching level of college teachers.
Answer: Thanks for your comments! We have revised as:
The budget constraints and measurement costs from the actual situation of the evaluation of teaching level of college teachers are set.
- The following sentences are too long for readers to easily follow because they convey too many meanings. I suggest these sentences should be separated into several sentences.
- Examining the influence of these factors on the teaching level of college teachers can help to accurately evaluate whether or not a teacher's teaching effect is good, and the level is high, and it also helps to understand the overall teaching level of a teacher from multiple aspects better and compare the differences between the teaching levels of different teachers fully.
Answer: Thanks for your comments! We have revised it in 1 Introduction as:
Examining the influence of these factors on the teaching level of college teachers can help to accurately evaluate whether or not a teacher's teaching effect is good, and the level is high. It also can help to understand the overall teaching level of a teacher from multiple aspects better and compare the differences among the teaching levels of different teachers fully.
- It is necessary to pay attention to these influencing factors on the objects of measurement and constructing different generalizability designs to reflect the relationship among them in the analysis process using generalizability theory, otherwise they may become hidden facets thereby exaggerating the generalized coefficient of the evaluation (Brennan, 2001a).
Answer: Thanks for your comments! We have revised in 1 Introduction as:
It is necessary to pay attention to these influencing factors on the objects of measurement and constructing different generalizability designs to reflect the relationship among them in the analysis process using the generalizability theory. Otherwise, these influencing factors may become hidden facets thereby exaggerating the generalizability coefficient of the evaluation (Brennan, 2001a).
- In Table 10, ns and ni indicate the total number of the optimal students and projects needed to evaluate the 10 teachers, n's is the average number of the optimal students needed to evaluate per teacher, n'i indicates the average number of optimal topics contained in the dimensions.
Answer: Thanks for your comments! We have revised it in 5 Discussion as:
In Table 10, and indicate the total number of the optimal students and items needed to evaluate the 10 teachers. In Table 10, is the average number of the optimal students needed to evaluate per teacher, indicates the average number of optimal items contained in the dimensions.
- The author mentioned that the Cauchy-Shiya inequality method and the LaGrange multiplier method were proposed to improve the discrete optimization method. The author should explain why the LaGrange multiplier method was used in stead of the Cauchy-Shiya inequality method in this study.
Answer: Thanks for your comments! We add one paragraph in 1 Introduction as follows:
The research of Li and Ou (2020) shows that the performance of Lagrange multiplier method is better than that of Cauchy Schwartz inequality method. Therefore, this study only discusses the Lagrange multiplier method.
- A research question should be proposed based on the research purpose.
Answer: Thanks for your comments! We add one paragraph in 1 Introduction as follows:
The purpose of this study is as follows: (1) Based on generalizability theory combined with the budget constraints, we unified the Lagrange multiplier formula and derived the optimal sample size estimators for different generalizability designs. (2) According to the optimal sample size estimators for different generalizability designs, we estimate the optimal number of students and the optimal number of items for the actual teaching level evaluation of college teachers, hoping to provide references for research similar to the evaluation of the teaching level of college teachers.
- This word “research” is generally uncountable. If the word becomes countable, that will mean research in different fields/areas. Please make sure the wording is correct in the study.
Answer: Thanks for your comments! We change the “researches” into the “research”.
- Because it is the main research instrument in this study, more descriptions of the Teaching Level Evaluation Questionnaire for College Teachers should be provided.
Answer: Thanks for your comments! We add one paragraph in 3.1 Data Collection as follows:
The questionnaire includes three dimensions (i. e. teaching method, teaching content, and teaching effect). The teaching method is the general term of the ways and means used by teachers and students in the teaching process to achieve the common teaching goals and complete the common teaching tasks.The teaching content is the main information that is intentionally transmitted in the process of interaction between learning and teaching, generally including curriculum standards, textbooks and courses. The teaching effect refers to the effect achieved by teachers through teaching behavior to guide students to acquire knowledge, such as better academic performance and greater progress made by students. Each dimension has 18 items, and there are 54 items in total on a 5-point scale (from 1 = Disagree at all to 5 = Agree very much). The Cronbach’s ɑ coefficient of the whole questionnaire is 0.88. The internal consistency coefficients of all dimensions and the questionnaire are 0.85, 0.80, 0.85 and 0.94. The correlation between the scores of all dimensions and the total score of the questionnaire was 0.62~0.78. We performed a series of Confirmatory Factor Analyses (CFA; Geiser, 2012) to identify the dimensions of the scales using a pilot sample data (n = 501) collected in 2020, prior to our formal research. Results indicated three dimensions (and each of these dimensions was examined using 18 items and averaged): teaching method (e. g., “the teacher is good at using multi-media, such as lantern slide, models, films, for teaching; the teacher summarizes and emphasizes the key points and difficult points clearly.”), teaching content (e.g., “the teacher introduces us the present trend of the subject and the background of the learning content; the teacher links learning content to practical life.”), and teaching effect (e.g., “through the study, I grasp the basic principles and theories of the curriculum; through the study, I learn how to solve problems by searching and using information resources.”). The model fit was acceptable (Schermelleh-Engel et al., 2003) and the validity of the questionnaire met the measurement requirements (χ2/df = 3.283, CFI = 0.927, TLI = 0.918, RMSEA = 0.066, SRMR = 0.039).
- The author mentioned that there were three dimensions in the questionnaire. Each dimension consisted of 18 questions, and there were 54 titles in total. It is suggested that the author should provide two or three sample questions in each item. In addition, what do those titles refer to?
Answer: Thanks for your comments! We give two or three sample questions in 3.1 Data Collection as following:
teaching method (e. g., “the teacher is good at using multi-media, such as lantern slide, models, films, for teaching; the teacher summarizes and emphasizes the key points and difficult points clearly.”), teaching content (e.g., “the teacher introduces us the present trend of the subject and the background of the learning content; the teacher links learning content to practical life.”), and teaching effect (e.g., “through the study, I grasp the basic principles and theories of the curriculum; through the study, I learn how to solve problems by searching and using information resources.”).
The titles is revised as items.
- It’s better to report the reliability coefficient of the overall questionnaire first and then the reliability coefficients of the three dimensions.
Answer: Thanks for your comments! We report the reliability coefficient of the overall questionnaire first and then the reliability coefficients of the three dimensions in 3.1 Data Collection as following:
The Cronbach’s ɑ coefficient of the whole questionnaire is 0.88.
- Please explain the following sentence more because it looks confusing. “The average correlation between the scores of all dimensions and the total score of the questionnaire was 0.82.”
Answer: Thanks for your comments! The sentence is revised as:
The correlation between the scores of all dimensions and the total score of the questionnaire was 0.62~0.78.
- The author mentioned that Table 1 shows the distribution of the number of students and types of students participating in the evaluation of the teaching level of college teachers. I cannot tell any type of students in Table 1.
Answer: Thanks for your comments! Table 1 is revised as:
Table 1 shows the number of teachers and students for different major type participating in the evaluation of the teaching level for college teachers.
Table 1 the number of students of the teacher for different major type in the evaluation
|
Teacher ID |
1 |
2 |
3 |
4 |
5 |
6 |
7 |
8 |
9 |
10 |
|
Student number |
43 |
39 |
42 |
39 |
39 |
37 |
38 |
42 |
39 |
39 |
|
Major type |
A |
S |
A |
E |
E |
A |
A |
S |
E |
S |
Note: A= Liberal arts; S= Science; E= Engineering
- How were the 10 teachers selected to participate in this study? What sampling method was adopted in this study? Were they at the same college?
Answer: Thanks for your comments! We add one paragraph in 3.2 Procedures as follows:
In total, we investigated 10 teachers (and 10 classes or 10 courses ) from three colleges and their teaching performance. All the 10 courses were mandatory, and had the same workload (one 45- minute lesson a week). Data were collected during the class in 45 minutes using a paper/pencil version survey administered to all students in these classes, first at the end of the first semester (T1, before the final exam; fall semester) and then at the beginning of the second semester (T2, spring semester). Research staff were trained before they administered the survey. Student assent was obtained, and this study received approval documents from the targeted university’s research ethics board (Institutional Review Board).
- It is suggested that the author should define the four factors (i.e. students, topics, dimensions and occasions) in the section of introduction or methods.
Answer: Thanks for your comments! The four factors (i.e. students, topics, dimensions and occasions) are defined in 3.3 Generalizability Design as following:
Among them, t denotes the evaluated teacher, i is the evaluated item, s denotes the evaluated student, v represents the dimension of the evaluation item (dimension or veidoo), and o denotes the evaluated occasion (occasion).
- Pedagogical implications should be provided based on the findings obtained in this study.
Answer: Thanks for your comments! We add one paragraph in 5 Discussion as follows:
Under budget constraints, the number of students can be reduced, even to about 30. If the evaluation occasion or time is appropriate, the questionnaire evaluation can also ensure considerable reliability. So, this gives us two inspirations: on the one hand, it is not necessary to implement large-scale questionnaire evaluation. A considerable number of students can also ensure the reliability of the evaluation, which saves the human, material and financial resources of colleges and universities; on the other hand, the evaluation occasion or time can be adjusted according to the actual situation, and the number of students and items in the evaluation can be appropriately reduced to the optimal sample size of students as 31 for each teacher and the optimal sample size of items as 7 for each dimension. It is not necessary to carry out large-scale questionnaire evaluation to obtain enough students and items, which is usually seen as more insurance. In fact, if the occasion or time is appropriate, a certain number of students and items will be enough.
- The reference list should be updated.
Answer: Thanks for your comments! We add some literatures as follows:
Li, G., & Ou, X. (2020). Comparing of LaGrange multiplication and Cauchy-Schwarz inequality for optimal sample size estimation. Statistics & Decision, 552(12), 29–33.
Geiser, C. (2012). Data analysis with Mplus. Guilford Press.
Schermelleh-Engel, K., Moosbrugger, H., & Müller, H. (2003). Evaluating the fit of structural equation models: Tests of significance and descriptive goodness-of-fit measures. Methods of Psychological Research Online, 8(2), 23–74.
Clayson, P. E., Carbine, K. A., Baldwin, S. A., Olsen, J. A., & Larson, M. J. (2021). Using generalizability theory and the erp reliability analysis (era) toolbox for assessing test-retest reliability of erp scores part 1: Algorithms, framework, and implementation. International Journal of Psychophysiology, 166, 174–187.
Vispoel, W. P., Xu, G., & Kilinc, M. (2020). Expanding G- Theory models to incorporate congeneric relationships: Illustrations using the big five inventory. Journal of Personality Assessment, 103(1), 429–442.
Li, G. (2019a). Psychological measurement. Beijing, China: Tsinghua Universiy Publishing House.
Li, G. (2019b). Estimation of optimal sample size under budget constraints for generalization theory based on LaGrange multiplier method. Statistics & Decision, 527(11), 13–16.
Zhang, J., & Lin, C. K. (2016). Generalizability theory with one-facet nonadditive models. Applied Psychological Measurement, 40(6), 367–386.
Zheng, F., & Gao, X. (2018). A sufficient condition for conditional extreme value in using lagrange multiplier method. Studies in College Mathematics, 21(2), 41–43.

Reviewer 2 Report
Dear Authors,
Thank you for entrusting me to review your article. Congratulations, you have written an interesting article about the teacher evaluation system in high schools. Overall, this article is good, but there are some notes that need to be improved, including;
Line 165: The purpose of this research is explained clearly again, if it can be written per item it will be clearer and easier for readers to follow. It is also necessary to describe the inferences from the results of this study.
Line 294: Add an explanation, of whether this questionnaire was developed by yourself (If so, what model or theory was used for developing the questionnaire). If adopted, from other sources, state the source and how about its validity and reliability.
Line 298: It would be clearer if we could add a description of what indicators of these dimensions are.
Line 307: It will be easier to understand if given a title for each content in the table, for example 43, 39, and so on What is it called.
Line 309: Why are students from this class taken, why not take a language class (for example). Maybe it would be better if it could be justified..
Line 318: This data analysis tool is developed independently or in software applications that have been developed by previous researchers. Described a few characteristics of the program.
Line 411: For each subtopic (item below) it is necessary to discuss the findings of this study with the findings of several relevant previous studies. Include some previous research results for each sub topic.
Line 504: Besides the limitations of this study or research, it is also necessary to explain what is the inferential of the results of this study.

Author Response
Thank you for entrusting me to review your article. Congratulations, you have written an interesting article about the teacher evaluation system in high schools. Overall, this article is good, but there are some notes that need to be improved, including;
Line 165: The purpose of this research is explained clearly again, if it can be written per item it will be clearer and easier for readers to follow. It is also necessary to describe the inferences from the results of this study.
Answer: Thanks for your comments! We add one paragraph in 1 Introduction as follows:
The purpose of this study is as follows: (1) Based on generalizability theory combined with the budget constraints, we unified the Lagrange multiplier formula and derived the optimal sample size estimators for different generalizability designs. (2) According to the optimal sample size estimators for different generalizability designs, we estimate the optimal number of students and the optimal number of items for the actual teaching level evaluation of college teachers, hoping to provide references for research similar to the evaluation of the teaching level of college teachers.
Also, we describe the inferences from the results of this study in 5 Discussion as follows:
Under budget constraints, the number of students can be reduced, even to about 30. If the evaluation occasion or time is appropriate, the questionnaire evaluation can also ensure considerable reliability. So, this gives us two inspirations: on the one hand, it is not necessary to implement large-scale questionnaire evaluation. A considerable number of students can also ensure the reliability of the evaluation, which saves the human, material and financial resources of colleges and universities; on the other hand, the evaluation occasion or time can be adjusted according to the actual situation, and the number of students and items in the evaluation can be appropriately reduced to the optimal sample size of students as 31 for each teacher and the optimal sample size of items as 7 for each dimension. It is not necessary to carry out large-scale questionnaire evaluation to obtain enough students and items, which is usually seen as more insurance. In fact, if the occasion or time is appropriate, a certain number of students and items will be enough.
Line 294: Add an explanation, of whether this questionnaire was developed by yourself (If so, what model or theory was used for developing the questionnaire). If adopted, from other sources, state the source and how about its validity and reliability.
Answer: Thanks for your comments! We add one paragraph in 3.1 Data Collection as follows:
The questionnaire includes three dimensions (i. e. teaching method, teaching content, and teaching effect). The teaching method is the general term of the ways and means used by teachers and students in the teaching process to achieve the common teaching goals and complete the common teaching tasks.The teaching content is the main information that is intentionally transmitted in the process of interaction between learning and teaching, generally including curriculum standards, textbooks and courses. The teaching effect refers to the effect achieved by teachers through teaching behavior to guide students to acquire knowledge, such as better academic performance and greater progress made by students. Each dimension has 18 items, and there are 54 items in total on a 5-point scale (from 1 = Disagree at all to 5 = Agree very much). The Cronbach’s ɑ coefficient of the whole questionnaire is 0.88. The internal consistency coefficients of all dimensions and the questionnaire are 0.85, 0.80, 0.85 and 0.94. The correlation between the scores of all dimensions and the total score of the questionnaire was 0.62~0.78. We performed a series of Confirmatory Factor Analyses (CFA; Geiser, 2012) to identify the dimensions of the scales using a pilot sample data (n = 501) collected in 2020, prior to our formal research. Results indicated three dimensions (and each of these dimensions was examined using 18 items and averaged): teaching method (e. g., “the teacher is good at using multi-media, such as lantern slide, models, films, for teaching; the teacher summarizes and emphasizes the key points and difficult points clearly.”), teaching content (e.g., “the teacher introduces us the present trend of the subject and the background of the learning content; the teacher links learning content to practical life.”), and teaching effect (e.g., “through the study, I grasp the basic principles and theories of the curriculum; through the study, I learn how to solve problems by searching and using information resources.”). The model fit was acceptable (Schermelleh-Engel et al., 2003) and the validity of the questionnaire met the measurement requirements (χ2/df = 3.283, CFI = 0.927, TLI = 0.918, RMSEA = 0.066, SRMR = 0.039).
Line 298: It would be clearer if we could add a description of what indicators of these dimensions are.
Answer: Thanks for your comments! We add some sentences in 3.1 Data Collection as follows:
The teaching method is the general term of the ways and means used by teachers and students in the teaching process to achieve the common teaching goals and complete the common teaching tasks.The teaching content is the main information that is intentionally transmitted in the process of interaction between learning and teaching, generally including curriculum standards, textbooks and courses. The teaching effect refers to the effect achieved by teachers through teaching behavior to guide students to acquire knowledge, such as better academic performance and greater progress made by students.
Line 307: It will be easier to understand if given a title for each content in the table, for example 43, 39, and so on What is it called.
Answer: Thanks for your comments! Table 1 is revised as:
Table 1 shows the number of teachers and students for different major type participating in the evaluation of the teaching level for college teachers.
Table 1 the number of students of the teacher for different major type in the evaluation
|
Teacher ID |
1 |
2 |
3 |
4 |
5 |
6 |
7 |
8 |
9 |
10 |
|
Student number |
43 |
39 |
42 |
39 |
39 |
37 |
38 |
42 |
39 |
39 |
|
Major type |
A |
S |
A |
E |
E |
A |
A |
S |
E |
S |
Note: A= Liberal arts; S= Science; E= Engineering
Line 309: Why are students from this class taken, why not take a language class (for example). Maybe it would be better if it could be justified.
Answer: Thanks for your comments! See the 3.2 Procddures: In total, we investigated 10 teachers (and 10 classes or 10 courses ) from three colleges and their teaching performance. All the 10 courses were mandatory, and had the same workload (one 45- minute lesson a week). Data were collected during the class in 45 minutes using a paper/pencil version survey administered to all students in these classes, first at the end of the first semester (T1, before the final exam; fall semester) and then at the beginning of the second semester (T2, spring semester). Research staff were trained before they administered the survey. Student assent was obtained, and this study received approval documents from the targeted university’s research ethics board (Institutional Review Board).
Line 318: This data analysis tool is developed independently or in software applications that have been developed by previous researchers. Described a few characteristics of the program.
Answer: Thanks for your comments! Analyses are performed in the urGENOVA software (Brennan, 2001b). Some programs were completed by writing the control card in the urGENOVA software.
Line 411: For each subtopic (item below) it is necessary to discuss the findings of this study with the findings of several relevant previous studies. Include some previous research results for each sub topic.
Answer: Thanks for your comments! To connect with some previous research results for each sub topic. We add some sentence in 5 Discussion such as:
From Table 9, these facets gradually emerge from the hidden facets with the facets added one by one, which results in that the “fixed” variance in the global fractional is gradually “liberated” (Brennan, 2001a). This also shows that these "liberated" facets may have a certain impact on the evaluation of the teaching level of college teachers, which is consistent with the research of Meyer et al. (2014).
From the analysis, this occasion factor is very important, which is consistent with the research of Iqbal et al. (2016).
Especially, when students are under the pressure of the exam, they have to make high scores on the teacher's evaluation (Vaillancourt, 2013; Iqbal et al., 2016).
Line 504: Besides the limitations of this study or research, it is also necessary to explain what is the inferential of the results of this study.
Answer: Thanks for your comments! We add one paragraph in 5 Discussion as follows:
Under budget constraints, the number of students can be reduced, even to about 30. If the evaluation occasion or time is appropriate, the questionnaire evaluation can also ensure considerable reliability. So, this gives us two inspirations: on the one hand, it is not necessary to implement large-scale questionnaire evaluation. A considerable number of students can also ensure the reliability of the evaluation, which saves the human, material and financial resources of colleges and universities; on the other hand, the evaluation occasion or time can be adjusted according to the actual situation, and the number of students and items in the evaluation can be appropriately reduced to the optimal sample size of students as 31 for each teacher and the optimal sample size of items as 7 for each dimension. It is not necessary to carry out large-scale questionnaire evaluation to obtain enough students and items, which is usually seen as more insurance. In fact, if the occasion or time is appropriate, a certain number of students and items will be enough.

Reviewer 3 Report
The topic is of interest if properly focussed, and indeed there has been literature on the topic since the 90s.
It is unclear how the authors recall Lagrangian multiplier techniques to optimize multiple variable problems that require differentiability and thus continuity of the variables involved to a case where the variables take discrete (natural) values.
On the other hand, Joseph-Louis Lagrange (born Giuseppe Luigi Lagrangia or Giuseppe Ludovico De la Grange Tournier, also reported as Giuseppe Luigi Lagrange or Lagrangia, was an Italian mathematician, later naturalized French, and internationally he is commonly referred as Lagrange, never have seen named in the way the authors do - they should revise their sources.
The paper includes some statistics formulas and data, but it is fuzzy how Lagrange multipliers are used. and how the results announced in the Abstract are deduced.
Author Response
The topic is of interest if properly focussed, and indeed there has been literature on the topic since the 90s.
It is unclear how the authors recall Lagrangian multiplier techniques to optimize multiple variable problems that require differentiability and thus continuity of the variables involved to a case where the variables take discrete (natural) values.
Answer: Thanks for your comments! See the 3.1 Data Collection: Each dimension has 18 items, and there are 54 items in total on a 5-point scale (from 1 = Disagree at all to 5 = Agree very much). In this study, because it is a 1-5 grade score, it is approximately regarded as a continuous variable rather than a discrete variable, so it is suitable for Lagrangian multiplier. The following documents can be referred to for this practice:
Meyer, P. J., Liu, X., & Mashburn, A. J. (2014). A practical solution to optimizing the reliability of teaching observation measures under budget constraints. Educational and Psychological Measurement, 74(2), 280–291.
Li, G. (2019b). Estimation of optimal sample size under budget constraints for generalization theory based on LaGrange multiplier method. Statistics & Decision, 527(11), 13–16.
Li, G., & Ou, X. (2020). Comparing of Lagrange multiplier and Cauchy-Schwarz inequality for optimal sample size estimation. Statistics & Decision, 552(12), 29–33.
On the other hand, Joseph-Louis Lagrange (born Giuseppe Luigi Lagrangia or Giuseppe Ludovico De la Grange Tournier, also reported as Giuseppe Luigi Lagrange or Lagrangia, was an Italian mathematician, later naturalized French, and internationally he is commonly referred as Lagrange, never have seen named in the way the authors do - they should revise their sources.
Answer: Thanks for your comments! We have revised the “LaGrange” as the “Lagrange” for this paper. thank you!
The paper includes some statistics formulas and data, but it is fuzzy how Lagrange multipliers are used. and how the results announced in the Abstract are deduced.
Answer: Thanks for your comments! Lagrange multiplier is to solve the optimization problem with equality constraints by introducing Lagrange multipliers (new scalar unknowns) in mathematical optimization problems. Let the given function of two variables z=ƒ (a, b) and additional conditions φ (a, b)=0, in order to find the extreme point of z=ƒ (a, b) under additional conditions, first do the Lagrangian function, where λ is a parameter.
Let F (a, b, λ) for a and b and λ of the first partial derivation be equal to zero, that is
F'a=ƒ'x(a,b)+ λφ' x(a,b)=0
F'b=ƒ'y(a,b)+ λφ' y(a,b)=0
F' λ=φ (a,b)=0
We can get A, b and λ. The (a, b) thus obtained is the function z=ƒ (a, b) under additional conditions φ (a, b)=0. If there is only one such point, it can be directly determined from the actual problem.
The Lagrange multiplier is a method of finding the extreme of a multivariate function whose variables are limited by one or more conditions. It can solve the problem of the optimization with equality constraints by introducing Lagrange multiplier (Wang & Wu, 1999). When using the Lagrange multiplier to solve a problem, the unified formula of Lagrange function is formulated as follows:
(1)
Where is the Lagrange function, andare unknown parameters of the function, is a new unknown scalar. Equation (1) can be interpreted as solving the extremism of the function under the constraint of the function by introducing a new unknown scalar (Zheng & Gao, 2018).
According to formula (1), if the teaching level evaluation of college teachers involves evaluation student-s, evaluation item-I, evaluation dimension-v and evaluation occasion (o, number of times), the Lagrange function can be expressed as:
(2)
Where represents the number of evaluation students, is the number of evaluation items, represents the dimension of the evaluation items, is the number of evaluation times, represents the relative error variance, is the new unknown scalar, c represents the cost of a single question, B indicates the budget for completing an evaluation. Formula (2) set the function and the additional condition, introducing the unknown parameterthereby finding the extreme value of the function under the restriction of additional conditions. indicates the actual cost is less than or equal to the budget. When, the maximum value of is obtained since c and B are fixed values. Therefore, formula (2) takes the extreme value of the relative error variance by introducing the Lagrange multiplier as the actual cost is not greater than the budget.
The following are some formulas for calculating the partial derivative of three generazability design:
1、(s:c)×i设计
2、(s:c)×(i:v)设计
3、(s:c)×(i:v)×o设计
According to the above formulas, the optimal sample size of all three designs can be estimated.

Reviewer 4 Report
I read the submitted article with pleasure. I have only two observations about him:
1. I miss a greater connection with the magazine. I am aware that this is a special issue. However, the word sustainability did not appear in the entire text. For this, a chapter needs to be added.
2. Literary sources. Many of them are not current. I therefore recommend revision and expansion with current articles.
Author Response
I read the submitted article with pleasure. I have only two observations about him:
- I miss a greater connection with the magazine. I am aware that this is a special issue. However, the word sustainability did not appear in the entire text. For this, a chapter needs to be added.
Answer: Thanks for your comments! We add one paragraph in 1 Introduction as follows:
Education is sustainable. If we want to maintain the sustainable development of higher education, we must have strong teachers and attach importance to college teachers’ teaching level. In this view, we should strengthen the supervision of teachers' teaching behavior for the reason that the evaluation is very important, but it needs to be scientific and reasonable.
- Literary sources. Many of them are not current. I therefore recommend revision and expansion with current articles.
Answer: Thanks for your comments! We add some literatures as follows:
Li, G., & Ou, X. (2020). Comparing of LaGrange multiplication and Cauchy-Schwarz inequality for optimal sample size estimation. Statistics & Decision, 552(12), 29–33.
Geiser, C. (2012). Data analysis with Mplus. Guilford Press.
Schermelleh-Engel, K., Moosbrugger, H., & Müller, H. (2003). Evaluating the fit of structural equation models: Tests of significance and descriptive goodness-of-fit measures. Methods of Psychological Research Online, 8(2), 23–74.
Clayson, P. E., Carbine, K. A., Baldwin, S. A., Olsen, J. A., & Larson, M. J. (2021). Using generalizability theory and the erp reliability analysis (era) toolbox for assessing test-retest reliability of erp scores part 1: Algorithms, framework, and implementation. International Journal of Psychophysiology, 166, 174–187.
Vispoel, W. P., Xu, G., & Kilinc, M. (2020). Expanding G- Theory models to incorporate congeneric relationships: Illustrations using the big five inventory. Journal of Personality Assessment, 103(1), 429–442.
Li, G. (2019a). Psychological measurement. Beijing, China: Tsinghua Universiy Publishing House.
Li, G. (2019b). Estimation of optimal sample size under budget constraints for generalization theory based on LaGrange multiplier method. Statistics & Decision, 527(11), 13–16.
Zhang, J., & Lin, C. K. (2016). Generalizability theory with one-facet nonadditive models. Applied Psychological Measurement, 40(6), 367–386.
Zheng, F., & Gao, X. (2018). A sufficient condition for conditional extreme value in using lagrange multiplier method. Studies in College Mathematics, 21(2), 41–43.

Round 2
Reviewer 1 Report
The author has made all changes suggested by myself and I feel that it is fit for publication. I commend the author on the fine contribution.
Author Response
The author has made all changes suggested by myself and I feel that it is fit for publication. I commend the author on the fine contribution.
Answer: thank you.
Reviewer 3 Report
The paper is poorly written with many repetitions of content and Grammarly with "First of all, first, second, secondl y,"and so on in the same paragraph (see to start end of the first page and beginning of the second page). That happens all over the text.
The Abstract Abstract: Education is sustainable. If we want to maintain the sustainable development of higher education, we must have strong teachers and attach importance to college teachers’ teaching level. 397 students are required to evaluate 10 teachers’ teaching level using the Teaching Level Evaluation Questionnaire for College Teachers, and we make different generalizability designs..” is not an Abstract. It is just two initial statements that do not give insight into the real content of the paper and the next sentence just reflects that the author has applied known methods on generalizability, it is an attempt to apply generalizability principles to a given case. There is no apparent new gain compared to the existing literature.
In their answer to the reviewer, the authors explain how the Lagrange multiplier works. That is unnecessary as it is a basic topic in any first calculus year at university. The main concern is that the Lagrangian method requires finding gradients, that is partial derivatives, just to get the necessary conditions. In this case, variables are discrete and it is unclear that taking the nearest integer, one gets an optimal solution.
If one wants to calculate the minimum of 3 + (x-22)(x-30,1)(x-32) subject to some condition may get 30,1 approx = 30 as a minimum (local) but quite likely the real minimum might be a number smaller than 21. The author does not mention any sufficient condition. The author provides 3 references, two of them being from his previous papers, where this kind of handling of Lagrangian multipliers for discrete variables, has been used.
The author tries to minimize the relative error variance, but does not explain what errors they refer to, the variable nv is unclearly stated what it represents.
Calculus is done in Eqs (5) and (6) with some estimated variances,… no consideration of error is done either.
